# "If he returns, receive him because he has realized that he needs assistance": A qualitative study exploring preferences for retention on antiretroviral therapy support in Malawi among Lighthouse Clinic clients

Gillian O'Bryan[1,2]*, Jacqueline Huwa[3], Odala Sande[3], Agness Thawani[3], Astrid Berner-Rodoreda[4], Hannock Tweya[1,2], Christine Kiruthu-Kamamia[2,3], Geldert Chiwaya[3], Caryl Feldacker[1,2]

1 Department of Global Health, University of Washington, Seattle, Washington, United States of America, 2 International Training and Education Center for Health (I-TECH), Seattle, Washington, United States of America, 3 Lighthouse Trust, Lilongwe, Malawi, 4 Heidelberg Institute of Global Health, University of Heidelberg, Heidelberg, Germany

* gilliano@uw.edu

## Abstract

### Introduction

Retention of people living with HIV (PLHIV) on antiretroviral therapy (ART) is critical. Retention is not static: clients cycle in and out of care over the course of treatment. Retention support is also evolving, with interventions like mobile health (mHealth) gaining traction. We aimed to explore ART clients' perspectives and recommend strategies for improved retention support including a two-way-texting (2wT) mHealth intervention in two clinics in Malawi.

### Materials and methods

We conducted focus group discussions (FGDs) with PLHIV on ART receiving 2wT or standard of care (SoC) retention support to understand client perspectives. FGDs were audio-recorded in Chichewa, translated and transcribed into English. We used the RE-AIM implementation science framework to report findings on the reach, effectiveness, adoption, implementation, and maintenance potential of retention interventions considering individual, clinic/organization, and community factors. Through rapid qualitative analysis, we identified key themes and subthemes.

### Results

Ten FGDs were conducted with 89 ART clients (aged ≥18 years; 55% female): four FGDs among 2wT participants and six among SoC participants. Few differences were observed between 2wT and SoC clients. All clients appreciated accessing at

**Data availability statement:** Our complete transcripts contain data that is sensitive or includes identifying information. We also believe that the Malawian Ministry of Health would like the confidentiality of the participants protected in accordance with the consent agreement. Due to these concerns, we are unable to make the full transcripts available to a wider audience. We will make the transcripts available to fellow researchers or reviewers who complete a data sharing agreement. The edited transcripts will be available on a case by case basis after reviewing all materials for any potentially identifying details. Interested researchers may contact Jane Edelson jedelson@uw.edu, Regulatory Specialist at the UW, for access to the transcripts.

**Funding:** Research reported in this publication was supported by the Fogarty International Center of the National Institutes of Health under Award Number R33TW011658 (CF & HT). The content is solely the responsibility of the authors and does not necessarily represent the official views of the National Institutes of Health. The funders had no role in study design, data collection and analysis, decision to publish, or preparation of the manuscript.

**Competing interests:** The authors have declared that no competing interests exist.

least one form of support and recommended continuous assistance, regardless of age, sex, or duration on ART. Clients wished to be encouraged by other PLHIV to allay common fears. Groups discussed the need for intensified support to retain young clients or those out of care longer. Individual responsibility was identified as necessary, but insufficient, to improve retention: respondents desired a more positive clinic environment to encourage persistence in care. Fear of stigma, unintended status disclosure, and negative interactions with clinic staff prevent some clients from returning to care.

## Conclusion

PLHIV want differentiated, continuous retention support from other PLHIV coupled with a positive clinic environment to promote retention and reengagement in care. As differentiated retention support is expensive, consideration of cost, feasibility, and sustainability is needed.

## Introduction

In the previous decade, antiretroviral therapy (ART) coverage increased remarkably in sub-Saharan Africa (SSA): in East and Southern Africa, an estimated 84% of over 20 million people living with HIV (PLHIV) are on ART; 94% of those on ART are virally suppressed [1]. However, gaps remain: of all PLHIV, 78% have suppressed viral load, indicating gaps in ART uptake and retention. Retention on ART is crucial to improve the health outcomes of PLHIV including achievement and maintenance of suppressed HIV viral load [2]. ART programs in SSA face challenges with low retention on ART and high loss to follow-up (LTFU), leading to the growth of differentiated service delivery (DSD) focused on client-centered care and increased viral load suppression [3].

Retention is not static, PLHIV cycle in and out of HIV care over time for various individual reasons [4], with common barriers including stigma, limited access to clinics, long waiting times, poor treatment from healthcare workers, lack of effective retention counseling, mobility, or preference for traditional medicines [1]. Retention patterns and client preferences may change over time, in part driven by differences in clinical state (e.g., comorbidity, pregnancy) or socio-economic contexts (e.g., divorce, work constraints, distance to clinics, disclosure), necessitating diversity in retention service provision [5–7]. In response to client cyclical retention patterns, a similar diversity of DSD approaches for retention support may better help support clients throughout their ART life course, with more effort early in care (the first 6–12 months) when risk of LTFU is highest and ART refill visits more frequent [8,9].

In addition to more novel interventions for retention support, the timing of retention efforts also merits review. Most retention efforts are reactive, waiting for ART clients to miss a visit, and are costly, involving intensive active tracing via phone calls or home visits to find and return clients to care. Recently, mobile health (mHealth)

interventions, such as short-message service (SMS), have shown promise to improve visit or medication adherence [10], with potential to lower retention costs within large-volume public clinics in SSA [11–13]. There is growing interest in identifying evidence-based, mHealth strategies to engage clients in care [14], with emphasis on those interventions that reflect local priorities, contexts, and constraints [15].

In Malawi, retention on ART remains low both in the first year of treatment at 80% of adults retained alive on ART at 12 months after initiation and long-term at 54% at 5 years after ART initiation [16,17]. Lighthouse Trust (LT) works in partnership with the Malawi Ministry of Health (MoH) to provide a choice of differentiated retention support services alongside routine HIV care services. In 2019, LT developed a two-way-texting (2wT) mHealth retention support intervention in collaboration with the University of Washington's International Training and Education Center for Health (I-TECH), the MoH, and technology partner Medic. 2wT aims to resolve retention issues before they occur and improve retention data quality, thereby reducing the workload of health personnel, addressing resource scarcity, and reducing tracing delays [18,19]. Initial results from the quasi-experimental study found that 2wT significantly increased the probability of 12-month retention on ART [20].

In Malawi, timely exploration of ART clients' experiences, reflections, and suggestions for improved retention support may result in new interventions to reduce disruptions in care or shorten delays in returning to care. Therefore, our objectives are to 1) describe clients' perspectives on retention activities, gaining understanding on how, when, where, and why clients want to receive retention support; 2) elicit suggestions for strengthening retention support services over time; and 3) gain clients' input on 2wT, specifically, to inform decisions around continued 2wT optimization and scale-up.

## Materials and methods

The qualitative methods and results explored in this paper are part of a larger mixed-methods study using routine program data to explore HIV care engagement and retention among clients who started ART at Lighthouse. The findings from the quantitative component of this study will be reported separately.

### Setting

LT is Malawi's largest public provider of HIV-related care and works in partnership with the MoH to provide integrated HIV testing, treatment, and support services across Malawi. In Lilongwe district, LT operates two centers of excellence (CoEs) located at Kamuzu Central Hospital campus (LT KCH Clinic (LH-KCH)) and Bwaila Hospital (Martin Preuss Clinic (MPC)), providing free ART services to approximately 38,000 clients. ART clients attend clinic visits scheduled monthly for the first three months after ART initiation followed by a visit at 6 months and if adherent on ART, visits are reduced to every 3–6-months intervals, depending on drug supplies and individual clinical status including viral load suppression status.

### Lighthouse retention services

LT offers a range of differentiated retention support services to clients to enhance retention on ART (Table 1). LT offers two main types of services to ART clients: early retention support and late retention support. In brief, early retention support includes standard of care (SoC) and two-way texting (2wT) services – both are implemented before clients miss an appointment. For all clients, LT offers late retention support beginning from 14 days after a missed appointment using the back-to-care (B2C) tracing approach, starting with SMS or a phone call and then escalating to home visits if clients cannot be reached [21–26].

Other LT retention support services are offered to all clients at any time, including a call center where clients can receive clinical consultations; the Nurse-led community ART program (NCAP) with ART delivered via community support groups; and Ndife Amodzi that pairs a high-risk HIV client with a community member volunteer for intense emotional support via home visits.

**Table 1. Retention support services offered to recipients of HIV care at Lighthouse.**

| |
|---|
| **EARLY RETENTION: BEFORE A VISIT AND UP TO 13 DAYS AFTER A MISSED APPOINTMENT** |
| **ART Budd**y – Standard of Care (SoC) <br> **Buddies are** offered within the first 12–15 months of ART initiation. A peer-supporter ("ART buddy") is assigned or paired to an ART client, to provide counseling on adherence, disclosure, and reminds ART clients of their next clinic appointment through phone calls. If a client misses their appointment, the peer supporter calls the client up to three times within 13 days of the missed appointment, to ascertain the reason for missing the appointment and reschedule the appointment. If a client does not return within 14 days, he/she is referred to Back-to-Care for phone or field tracing. |
| **Two-way texting (2wT)** <br> Offered to new and existing ART clients who own a mobile phone and are willing to opt into the 2wT retention support service. Clients receive automated weekly motivational messages and visit reminder messages on days 3 and 1 before the clinic appointment via SMS. Visit reminder messages prompt participants to respond with a number: "1" for "yes" if they are attending the clinic visit, or "0" for "no" if they are not attending the visit. A "yes" response stops the reminders, while a "no" triggers automated messages to determine if the client wants to reschedule their appointment date, has transferred to another ART clinic, or has another non-clinical concern. If a client misses an appointment, he/she receives follow-up reminders on days 2, 5, or 11 after the missed visit. Messages are in Chichewa or English according to the participant's preference. If a client does not report to the clinic by 14 days, the 2wT system creates alerts for referral to Back-to-care for phone or field tracing. |
| **LATE RETENTON: ≥ 14 DAYS POST MISSED VISIT** |
| **Back to care (B2C)** <br> Designed to improve ART retention by bringing back clients who miss a clinic appointment by at least 14 days. In the B2C program, a team of tracers manually verifies the list of clients who miss appointments by ≥ 14 days, to rule out EMRS data errors and determine clients who consented for tracing. The B2C team traces clients through phone calls or home visits, using a standard tracing form. The B2C team updates the health information system with ART outcomes for successfully traced clients. |
| **Welcome Back culture** <br> Offered by treatment supporters to clients who interrupted ART upon re-engagement in care. A package of services (4 D approach) is offered which includes (1) Deflating the fears of returning to care so that clients feel welcomed, (2) Discussing the reasons for interrupting treatment, (3) Directing clients to appropriate services based on the reasons for interrupting treatment, (4) Decorate, by appreciating and acknowledging clients for treatment continuation and viral load suppression. |

## Participant selection and eligibility

We conducted 10 focus group discussions (FGDs) involving adults (aged ≥18 years) living with HIV and on ART to obtain a deeper understanding of the social and behavioral factors that affect clients' HIV care engagement in 2wT retention support groups (n = 4) and in standard care (n = 6). We selected focus groups rather than individual interviews to help spur discussion on social norms, values, expectations, and beliefs. Lively discussions elicited perspectives on the effectiveness of each current retention strategy, suggestions for improvement, and how to tailor approaches to client needs.

We used stratified purposeful sampling to recruit clients in the 2wT and SoC groups. FGD participants from the 2wT groups (n = 4) had received 2wT retention support for at-least 6 months. Within groups who had received 2wT, we further disaggregated recruitment to achieve a sample of participants who were (i) active, defined as interaction via the 2wT SMS platform on ≥15 occasions (this could be either clients replying to automated visit reminder messages, or live interactions with healthcare providers through SMS), and (ii) those who were inactive, defined as not responding to 2wT visit reminder, or responded not more than 3 times. Within the 6 FGDs conducted within the SoC category, criteria included ART status defined as (i) new on ART for 6–12 months (2 groups) or (ii) on ART > 1 year (4 groups). Among the 4 FGDs with PLHIV on ART > 1 year, 2 groups were held with those whose retention status was defined as (a) excellent retention (never missed clinic appointment since ART initiation) and 2 groups among those who (b) frequently missed visits (had at least 4 missed appointments) by at least 14 days.

For the selection process, we generated a complete list of eligible clients; LT retention team and expert clients called eligible clients and invited them to participate in FGDs via phone. Upon arriving at the facility, the study team provided eligible clients information via written material or read aloud, according to client preference and reading level. Recruitment procedures were conducted concurrent with data collection between March and April 2024. All individuals completed a written informed consent before participating in FGDs. A notetaker recorded group dynamics and non-verbal cues. A

highly trained qualitative researcher conducted FGDs at LT in Chichewa, audio-recorded, and translated and transcribed the FGD into English. Each FGD took approximately 1.5 hours.

## Theorical Framework

We include both the typical definitions of the components of the RE-AIM framework, as well as modifications to guide discussions with clients about RE-AIM dimensions at both the individual (i.e., those who are intended to benefit) and staff/setting-levels (how clients suggest staff or organizational improvements to retention support) [27–29].

Individual level:

- Reach is defined as the absolute number, proportion, and representativeness of individuals who are willing to participate in each initiative, intervention, or program.

- Effectiveness is defined as the perceived impact of retention support on individual-level outcomes, including potential negative effects, quality of life, and economic outcomes.

Staff and setting levels:

- Adoption is typically defined as the absolute number, proportion, and representativeness of the individuals who deliver the program – in this case, retention team members at LT. We solicited guidance from clients for how these healthcare workers can better adopt retention interventions to support clients.

- Implementation typically refers to the healthcare workers' fidelity to the various elements of an intervention's protocol, including consistency of delivery as intended. Client guidance for how LT and the various retention team members can better implement diverse retention interventions is critical to client acceptance of retention support.

- Maintenance is typically defined as the extent to which a program or policy becomes institutionalized or part of the routine organizational practices and policies. Client guidance for how these retention activities can be adapted for longer-term maintenance of retention acceptance over time is critical to consistent engagement of clients in these interventions, reducing resource wastage.

We extend the likelihood of applying RE-AIM findings to the client, clinic, and community contexts by considering the Practical Robust Implementation and Sustainability Model (PRISM) to translate results into actions [30].

## Data analysis

We used rapid qualitative analysis to identify key themes and subthemes from FGDs. Rapid qualitative analysis is an implementation science (IS) method used increasingly to generate rigorous, qualitative results in a shorter time compared to traditional qualitative methods [31] and is suitable for studies designed to inform practice, policy, or program adaptations in real-world settings [32,33]. Two investigators conducted the analysis for FGDs, each investigator independently condensed each FGD transcript into a 1–2-page transcript summary, organized by topic ("domain") with bullet points and key quotes that captured the essence of the transcript, discrepancies were compared and resolved through discussion between analysts. Members of the LT team reviewed and verified the final list of themes. We conducted a participant reflection meeting with a small group of expert PLHIV clients to help interpret results of preliminary findings and strengthen suggestions for improvement.

## Ethics

The Malawi National Health Sciences Research Committee (#23/10/4258) and the University of Washington, USA ethics review board (STUDY00101060) approved the study protocol. All FGD participants provided written informed consent.

## Results

All FGDs took place at Bwaila Hospital in Lilongwe but participants were from LH-KCH and MPC clinics. A total of 89 individuals participated across 10 FGDs conducted between March and April 2024 (Table 2). An average of nine individuals participated in each FGD (range 7–11). Overall, 40 males (45%) and 49 females (55%) participated.

We organized qualitative results using the RE-AIM implementation framework to help evaluate the reach, effectiveness, adoption, implementation, and maintenance potential of retention interventions with consideration of individual, clinic or organization, and community factors [27–29].

### Reach: How best to engage diverse clients

**Keep up visit reminders.** Across FGD SoC and 2wT groups, all participants acknowledged having accessed and appreciated retention support services provided by LT and the larger health system, *noting that "the facility has set in place good procedures of reminding us of an appointment… So, we need to appreciate them for all these good interventions in reminding us for an appointment date." (FGD 10).* Reminders were considered important for all clients, serving as a prompt to arrange transport or work schedules. Reminders should be the same for all.

> *"I think a patient is a patient, whether identified today, last year, or whatever year, they are all the same. The way one who started a long time ago can forget is the same as someone who is new, we are the same, I feel it should be the same and good." (FGD 9, Female)*

Although there was near universal support for appointment reminders using both SMS and phone, the order of the reminder support differed among the participants. Participants who received 2wT tended to include SMS first in the progression of escalated retention support, suggesting that *"for those who did not come after a text message, then a phone call will do"* (FGD 2, Female). Others felt either an SMS message or phone call could serve as the first retention support option, believing that *"the best way is to use the phones through calling or texting depending on the issues at hand." (FGD 2, Female)*

**Escalate retention interventions for clients with specific needs.** Participants in both the 2wT and SoC groups saw the need to intensify retention intervention methods when clients missed multiple visits, moving from SMS or phone followed by a home visit if previous attempts were unsuccessful in returning the client to care.

> *"And if you call him and is still not coming, we people left our physical addresses of where we stay. So, you are supposed to search for the person and find them, hear their problems and their reasons why they are not coming." (FGD 7, Male).*

**Table 2. Characteristics of focus group discussions.**

| No. | Group | Criteria | N | M-F ratio |
|---|---|---|---|---|
| 1 | 2wT | RCT active | 9 | 1:8 |
| 2 | | RCT inactive | 10 | 5:5 |
| 3 | | Quasi-experimental active | 10 | 6:4 |
| 4 | | Quasi-experimental inactive | 10 | 5:5 |
| 5 | SoC | On ART for 1 year with frequently missed visits | 7 | 5:2 |
| 6 | | | 9 | 3:6 |
| 7 | | New on ART (6–12 months) | 11 | 5:6 |
| 8 | | | 7 | 1:6 |
| 9 | | On ART 2–3 years with excellent retention | 9 | 5:4 |
| 10 | | | 7 | 4:3 |

The escalation of retention support was considered crucial to engage those who have extenuating circumstances, like sickness or frequent travel that could complicate attending visits. Also, as all clients knew that they provided addresses, it was expected that home visits would be conducted for those who failed to return to the facility after missing a clinic visit for a long period.

**Vary retention support delivery based on client characteristics.** Across 2wT and SoC groups, including across time on ART, feedback on what retention support was most needed or desired varied. For example, some participants noted that those newly initiated on ART might need more intensive support while they are in the process of accepting their HIV status, suggesting that new initiates might benefit from a home visit or more frequent contact.

*"For a new initiate, it takes time to get to the point where one has fully accepted their status and what they are supposed to be doing… So, a new initiate needs to be contacted frequently until she gets in line… if it's phone it's okay. You also feel good that other people remember you. To say the truth, you feel valued, and you know that your life is important." (FGD 7, Female)*

On the other hand, participants also noted that more established clients on ART might experience medication fatigue, suggesting the need to reinvigorate clients who have been on ART a long time.

*"You might say this one has been taking medication for long and knows everything. But they might take it for granted and say, 'I am tired of the medication, I want to stop, I am very tired.' … You must visit this person or call this person so that if they had ideas to stop the medication, they get encouraged and cannot stop taking medication." (FGD 8, Female)*

Participants also identified client age as a reason to differentiate support. Many thought that younger patients need more intensive support because they are more likely to forget their medication and are less "serious," suggesting the need for group or peer support. Respondents noted that a key challenge for younger, newly diagnosed HIV-positive clients is having to start medication when you are not sick.

*"To say the truth is that it is not an easy thing for one to accept when they are told that, 'you are HIV positive'…Very few of them are strong and will respond to say "aah it's okay, I will accept it and start taking medication." Most young people need this kind of conference, and someone should counsel them so that they should understand that, 'you will not die today when you start taking the drugs, this will make you to look good." (FGD 6, Male)*

Others said that older patients, including those with more advanced HIV or home-bound clients, have more issues with memory, requiring more involvement of guardians or more home visits.

*"So, I would recommend they should have a special reminder to the aged 60 years above as they have depleted their memory and are slow in acting. You may wish to remind these aged through their guardian… You may phone call them through the guardians." (FGD 10, Male)*

As these older or sicker patients could be unable to travel to the facility for ART refill visits, this additional attention would be warranted to help them retain in care.

**Stigma and fear HIV status disclosure still affect retention support preferences and choices.** Participants in both the 2wT and SoC groups expressed concerns that others can read SMS messages containing HIV information on a shared phone, unlike a phone call in which a healthcare worker can verbally verify the call recipient before discussing any information including HIV-related.

*"I feel like the information that is sent is enough, there is no need to add HIV related information. This is our secret. Even if we leave our phone and someone checks them, they will not be able to know what the messages are all about. If those messages have HIV related information, then it will make someone know your status." (FGD 1, Female)*

Participants cited fear of accidental disclosure as a reason why some might decline 2wT, and why 2wT motivational messages should not include HIV-related information.

### Effectiveness: how to improve SoC and 2wT retention support impact

**Encourage individual responsibility.** Several participants noted that for retention support to be effective, clients, themselves, should take personal responsibility for their health and ultimately their retention on ART as part of accepting one's HIV-positive status.

*"Firstly, you can never forget the appointment date when one has accepted that you have a problem. I have never missed an appointment date since I started. I accepted that I have a problem. For me to be healthy, I must be responsible… For my life to be better I must be serious and report to the clinic." (FGD 5, Male)*

From this responsibility and acceptance, participants also discussed that clients who are engaged in HIV self-management could relieve the burden on healthcare workers who are responsible for their support: "*I should make myself available* [for my visits] *because they are helping my life, not their life. So, we are supposed to encourage ourselves." (FGD 7, Female). Others* suggested that multiple reminders should not be necessary: *"It is not good that the same person should be reminded here and again, as an individual we just need to be responsible enough for our care." (FGD 10, Male)*

**Motivate clients with non-HIV related health information.** Current 2wT participants appreciated the motivational messages because they provide education on general health topics not related to HIV including hygiene and family planning, encouraging positive health decisions. *"They encourage us about our health. Sometimes they write messages that we have to remember to wash our hands. It is one way of encouraging us and I like that." (FGD 1, Female).* Participants also liked the messages about general health, wellness, and prevention of other illnesses.

*"It* [2wT] *helps to guide in our daily life operations, for example, cholera issues and house hygiene. Sometimes you may realize that your surrounding area is not tidied up, so we refer to the message while we are cleaning the surrounding area." (FGD 2, Male)*

Recommendations from current 2wT clients to improve motivational messages included including more seasonal messages (ex. to inform/educate during a cholera outbreak), and including religious messages to balance physical, emotional, and spiritual wellbeing.

*"Health goes together with spiritual and when you send the messages, you should include the verses, you should send the verses (in the messages) and we will interpret it ourselves." (FGD 4, Male)*

Participants felt the religious messages could pertain to any religion (e.g., Christian and Muslim) and the person receiving the message would know for whom the message is intended.

### Adoption: Client suggestions to increase retention support from healthcare workers

**Healthcare workers should interact more with clients to increase engagement.** In addition to appreciation for being reached with reminders and motivational messages, 2wT participants felt that interaction with the healthcare workers via SMS made them feel cared for.

*"When you tell them that you will not report to the clinic, they ask why, and when you tell them, they ask if they should change the date. That makes one feel cared for." (FGD 1, Female)*

Others felt that retention support, especially home visits, could benefit from more clinician participation to provide auxiliary health services like HIV testing for family members and care for those too sick to attend.

*"It should be the same nurse and doctors because if you get others like technicians then it will not help because the person/client may be suffering from some diseases. The doctors know what one is suffering from when they see them. It needs someone who was well trained, someone who graduated." (FGD 5, Male)*

However, other participants raised concerns about clinicians visiting clients at home due to stigma and accidental disclosure of HIV status as clinicians could, *"arrive at a home, you stop the car, and you are wearing your white attire, and they know you are from clinic."* (FGD 7-SoC new on ART-99). Participants suggested that clinicians should not conduct home visits in uniform and that healthcare workers could be met at a more anonymous community venue for anonymous support.

**Healthcare workers should adopt a more empathic and kind attitude.** Participants highlighted that retention is more than the medication, being significantly influenced by the quality of care and a respectful environment. Most believed LT staff was courteous and kind. From FGD 6, one respondent stated that they*, "would like to appreciate what the clinic does, when one reports to the clinic for to start treatment you welcome them warmly, in loving, caring and happy manner.*" Another reinforced the caring healthcare workers*, "The way you welcome us in this facility it's unique and this hospital is different with other hospital...I realized the workers here are good and they know what they are doing. This is really a calling for you*" (FGD 3). But not all FGD participants had such pleasant interactions. However, participants noted that not all providers had adopted an attitude that encouraged client retention support, showing a lack of empathy and respect for those facing retention challenges.

*"Some may shout at you and that can make one be scared or backslide as they are afraid of being shouted at, maybe they forgot and have more drugs remaining. So, they may be talked at in a bad way… That can make people who are new on ART to be scared… Sometimes when we see a provider in a room, we shun that room and go in another room and sometimes we make delays deliberate." (FGD 8, Female)*

In FGD 5, an exchange between participants was revealing. One of the participants spoke of the reception, noting that "They [reception staff] speak in an insulting way as if we are here for no good reason. *"Don't you know where you should stand? Go that side. Don't disturb me. Go that side*" (*mimicking the provide*r) (FGD5). But, in response, others chimed in to suggest a contrast with the respect and care from retention teams "*who make calls are excellent*", and "*we feel good on the phone.*"

### Implementation: Clients' suggestion for improved retention intervention implementation

**Expand peer-to-peer support.** Participants were of the view that it matters who provides retention support services. Retention support provided by other HIV-positive clients, like an ART *Buddy*, is most helpful, particularly because learning from the lived experiences of people who have been on ART is encouraging and can allay common fears and help those newly diagnosed HIV-positive accept their status. One participant talked about how their ART Buddy support was crucial to overcome ART side effects.

*"It reached a point where I was scared to take the medication in the evening, but the person encouraged me and said 'The drugs have not gotten used to your body and it will all stop. Don't worry about anything that you experience when you take the drugs. It will all stop once the body gets used [to the medication]'... Without that person then one would*

*think let me just sleep and not take the drugs. The advantage of that person is that they encourage us in some areas."*
*(FGD 9, Male)*

Retention support provided by other HIV-positive clients who are open about and have accepted their status can also serve as a visual example of what is possible in terms of the overall health and appearance of someone living with HIV when taking ARVs properly.

*"It is so nice to see someone who is also positive. It encourages one by just looking at another person's health/appearance, one can say I will be taking the medication "if this person looks like this and vice versa." If you meet people who look healthy, it gives you courage." (FGD 6, Male)*

**Consider gender in retention activities implementation.** Overwhelmingly participants thought that all retention methods (e.g., phone, SMS) should be available to both genders, and that frequency and intensity should not be different for reasons related to equity.

*"Firstly, we should look at who is the woman and who is the man. I am looking at 1, they are all human beings and have the same life, and they also have rights to live, 100%. I feel they should have the same support there is no reason whether one is a man or a woman, the treatment will be the same." (FGD 4, Male)*

However, they did consider that men and women may have different lifestyles that could influence retention support methods. For example, one male participant noted that men travel more away from home, suggesting that, "*you may not find us as we may be out. So, for me, I still think messages or calls are good.*" (FGD 3). Another male respondent noted that gender matching would be appropriate for home visits. One suggestion was that when tracers are conducting home visits, they should be in teams of two, a male and a female, to avoid issues related to gender or marital status. Or, if the tracer and client gender were opposite, providing advance notice for home visits would help clients to navigate potential issues with their partner there by preventing any issues that may arise due to partner suspicions.

*"They cannot come without asking for permission… So, one can choose if you feel like your spouse will not take it well. You can choose to meet somewhere." (FGD 6, Female)*

For counseling, some also noted a preference for same-sex counseling teams. One female respondent stated that:

*"If they are to come here to receive counseling, a man and a woman will not be able to understand each other. They will just be thinking about other things. But with a fellow woman, when you are telling you can also be having some interesting chat and be laughing, and you are understanding the content." (FGD 8, Female)*

Same-sex counseling pairs may also improve peer-to-peer counseling. One female participant reflected on her interaction with a female counselor after returning from an ART gap:

*"She encouraged me to the extent that I asked her, 'will my face look like the way you are looking?' She encouraged me a lot and she told me, 'Yes, if you just follow the schedule of taking the medication your face will get back,' (healed) because the rash was starting from the face to the whole body" (FGD 7, Female).*

### Maintenance: client recommendations to sustain retention gains

**Improve 2wT education and implementation.** More education on 2wT intervention components is needed to clarify how the messages work and how to interact with the healthcare workers. For 2wT participants, there was some confusion

over 2wT frequency and client costs, suggesting a need for improved 2wT awareness and education at enrollment. Some 2wT participants were unclear on the frequency of the weekly motivational message or that SMS delivery confirmation was not provided to confirm client responses. Lack of understanding of 2wT intervention details could lead to reduced 2wT interest or uptake as noted by one client who said, *"It shows that none of us can understand the entire text message process" (FGD 2, Male)*. One 2wT participant did not know that 2wT is free to send and receive texts across networks, stating that they do not respond to 2wT visit reminder messages because of perceived costs from their network provider (TNM vs. Airtel).

> *"Additionally, on the messages sent, most people fail to respond because the number that we are given for us to respond is TNM line, of which it charges when we respond through Airtel number." (FGD 4, Male)*

Moreover, early implementation weaknesses in the syncing of appointment dates between 2wT and the clinic records led some 2wT participants to receive missed visit reminders when they had already attended the visit. This caused confusion because, *"If I responded that I came for drug refill, I am surprised that the following day I would still receive a text message reminding me of the same." (FGD 2, Female)*.

Despite identified 2wT weaknesses, there was interest among SoC clients in joining the 2wT intervention, reinforcing the need for more awareness raised within the clinics to enroll interested and eligible clients. *"If it's starting today, I would say enroll me so that I should be receiving these messages." (FGD 7, Male)*

**Continue retention support for the lifetime of the client.** Participants felt that retention support should continue as long as the client is on ART because HIV is a chronic, lifelong disease.

> *"No, they* [retention support] *should not stop. Similarly, you encourage us because the medication is lifelong. We take them and reach a point where we get tired and think of stopping taking them. When you come or call us then we get courage that we shouldn't stop. That is why we continue taking medication because of people like you." (FGD 9, Female)*

> *"There is no end because the treatment is life long, unless if the buddy moves to another country, then you can look for another one, or maybe one dies then we part. If they are alive, there is no parting ways" (FGD 01, Female)*

Some participants noted that Buddy ART support should also be extended, from the current 12 months to at least 2 years, ending based on counsellor assessment or determination of retention progress. In contrast, clients expressed a desire for ongoing 2wT support without a predetermined end date.

**Disseminate more information to clients about existing retention support.** Lastly, clients suggested several retention interventions that already exist at LT, showing a lack of awareness of the diversity of support and services offered to clients. Clients suggested aligning clinic appointment dates with antenatal or post-natal care appointments for themselves or their children, which LT already tries to do if clients note other scheduled visits at the same facility or hospital campus. Other participants wanted ART pickup in smaller, community-based clinics, reflecting lack of awareness that LT already distributes ART to 7 community-based clinics in Lilongwe. Another person wanted to be able to call the clinic to report a delay in visit attendance, showing that the LT Clinic Hotline is not well known. While suggestions for providing phones or transport are outside the reach of routine retention services, it is noted that clients often lack resources to attend visits.

## Discussion

In this study we aimed to describe the perspectives of ART clients from LT's public clinics in Lilongwe, Malawi on various retention services. Participant responses were largely consistent irrespective of time on ART or current retention support (2wT or SoC). The RE-AIM framework helped structure findings on how, when, where, and why these ART clients

preferred different retention support services over time. This novel RE-AIM application could help ART programs adapt retention efforts more inclusively and equitably for diverse clients [34]. Consideration of multi-level factors (individual, clinic, and community) may also inform policy and practice improvements at LT to augment the reach, effectiveness, adoption, implementation, and maintenance of both 2wT and SoC retention efforts. [35].

At individual level, participants recognized their personal responsibility to stay healthy and adhere to ART [9,36], but still appreciated being reached by PLHIV as part of continuous retention support. Clients largely felt that retention services should reach clients equally irrespective of age, sex, or duration on ART. However, there was widespread understanding that some groups of clients (e.g., adolescents/young adults, older clients or those out of care for longer) may require more intensified or differentiated support. Evidence also suggests that gender-focused retention efforts are effective, with a large amount of evidence supporting gender-matched interventions for women and mother-baby pairs [37–40]. Other PLHIV peer support models improve retention among women in community settings [41], or for teens, including "teen clubs" for adolescent PLHIV in Malawi [42,43]. Men are often neglected in targeted retention efforts [8,44] but may benefit equally from positive clinic experiences or connections with other men who are successful on ART [45]. Therefore, to reach more clients, LT should consider ART *Buddies* and ART retention counseling matched by gender and age for clients who are new to ART, older, younger, or who frequently miss visits to reach and retain more clients.

To aid adoption of positive retention efforts at the provider and clinic levels, participants suggested that healthcare workers, including clinicians, should prioritize creating an empathetic and non-judgmental atmosphere of client acceptance, respect, and care. It is known that clients who feel ashamed or fear poor treatment may be more resistant to return [44,46], while clients who receive comfort or acknowledgement at the clinic maybe more likely to reengage [37,47]. However, a qualitative study among HCWs in South Africa found that providers largely understood client retention barriers, including fear of mistreatment, but the additional workload of returning clients to care influenced their treatment and lack of patience with clients [48]. LT could recommit to reducing negative clinic interactions between staff and clients, especially those who disengage from care, enhancing the likelihood of client retention and reengagement. More healthcare worker refresher training on language, tone, and volume coupled with supportive supervision could help ensure an environment of client-centered care. Efforts to reduce the workload associated with restarting clients on ART may also help providers adopt better attitudes.

To better implement retention support, participants in our study expressed the desire for varied, not static, retention activities. For clinic visit reminders or missed visit alerts, clients liked both SMS or phone calls. With support for either method, efficiency should be optimized. Costs are lower for the 2wT approach using SMS reminders over reminder phone calls [49], with savings to grow with mHealth implemented at scale [50,51]. As 2wT does not reach all clients [19], other complementary efforts, especially for harder to reach clients, will still be needed. For example, clients who remained out of care for longer periods might require home visits to return to care. Moreover, client retention education must increase. Clients may be overwhelmed at ART initiation when retention support options are discussed and have difficulty understanding or choosing the best retention support service options. To improve retention program implementation, LT should continuously disseminate options to engage or reengage in care via posters, recorded audio messages during client waiting periods, and retention team outreach to clients at the clinic.

Lastly, implementing a diversity of client-centered retention support services is labor-intensive and costly [52]. Offering a choice of differentiated retention support requires dedicated staff, awareness raising, quality monitoring and evaluation, and persistent efforts for those hardest to reach and retain. To ensure sustainability and maintenance of efforts, the Malawi MoH emphasizes equal consideration of cost, feasibility, and sustainability in program planning to maintain quality client retention support [53]. The global donor community must take notice with investments focusing on developing cost efficient, innovative strategies that align with clients' evolving needs and preferences and can be transitioned to and sustained by national health systems including mHealth interventions.

## Strengths and Limitations

The strengths of our study include diverse groups of FGD participants (2wT vs. SoC, new on ART vs. established in care, excellent retention vs. poor retention) which enriched the scope of our findings by sourcing feedback and opinions from relevant groups in a real-world, public ART program setting. In addition to this, using the RE-AIM framework provided a robust, systematic, and comprehensive exploration of the existing retention programs, creating opportunity for further refinement. However, due to the nature of our study, we could not completely avoid social desirability bias, as some participants' responses might have been influenced by other participants and been more favorable to LT programs as LT clients. Lastly, the constructs of the RE-AIM implementation framework are interrelated. Our organization of findings under RE-AIM constructs in this analysis is subject to our own interpretations.

## Conclusion

Clients want to be retained in care and are open about the support they need. A diverse menu of retention options, coupled with a supportive clinical environment, could be more effective to reduce retention gaps. Clients recognize their role in assuming ownership of their health, but healthcare workers need to adopt attitudes of openness and empathy to reengage clients in ART services after gaps. To better implement client-centered retention support, some differentiation by age, gender, or length of time on ART should be considered. To maintain these efforts, considerations of cost, feasibility, and sustainability are needed to ensure that changing clients' needs, preferences, and concerns are addressed swiftly. Retaining and reengaging more clients in ART care is a key component of creating an AIDS free generation.

## Acknowledgments

The authors thank the Malawi Ministry of Health for their dedication, ongoing support, and leadership to retention and reengagement programming for clients in ART care. The authors also thank the Lighthouse Trust clinical and programmatic staff for their tremendous dedication and skilled capacity to retention support services. Finally, authors would like to thank participants for their generosity in providing their time to participate in FGDs and invaluable insights into retention support preferences.

## Author contributions

**Conceptualization:** Jacqueline Huwa, Hannock Tweya, Caryl Feldacker.

**Data curation:** Odala Sande, Agness Thawani, Geldert Chiwaya.

**Formal analysis:** Gillian O'Bryan, Odala Sande, Hannock Tweya, Caryl Feldacker.

**Funding acquisition:** Caryl Feldacker.

**Investigation:** Caryl Feldacker.

**Methodology:** Gillian O'Bryan, Caryl Feldacker.

**Project administration:** Jacqueline Huwa.

**Supervision:** Jacqueline Huwa.

**Writing – original draft:** Gillian O'Bryan, Caryl Feldacker.

**Writing – review & editing:** Gillian O'Bryan, Jacqueline Huwa, Odala Sande, Agness Thawani, Astrid Berner-Rodoreda, Hannock Tweya, Christine Kiruthu-Kamamia, Geldert Chiwaya, Caryl Feldacker.

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
