## [Decision Letter · Decision Letter 0]

22 Dec 2024

PONE-D-24-35611“If he returns, receive him because he has realized that he needs assistance”: exploring cyclical retention with antiretroviral therapy clients at Lighthouse Clinic, MalawiPLOS ONE

Dear Dr. O’Bryan,

Thank you for submitting your manuscript to PLOS ONE. After careful consideration, we feel that it has merit but does not fully meet PLOS ONE’s publication criteria as it currently stands. Therefore, we invite you to submit a revised version of the manuscript that addresses the points raised during the review process.

We look forward to receiving your revised manuscript.

Kind regards,

Joseph KB Matovu, Ph.D.

Academic Editor

PLOS ONE

2. You have indicated that data is available from jedelson@uw.edu.  Please can we ask you to provide us with a general contact email address for the data requests, so readers can request access in perpetuity. If a general email is not available please provide a link to a website where readers can obtain access to data.

Additional Editor Comments:

1.        I did not see the quotation used in the study title. Such quotations, if any have to be used, should be based on the results shared in the body of the paper.

The authors need to ensure that the reporting of the findings follows standard qualitative reporting guidelines – e.g. COREQ, RATS, etc., and this should be well spelt out in the paper.The study seems to be on ways to improve client retention and has less, if any focus, on cyclical retention, although this is presented as the catch term in the study title. I think the study title can be refocused to something in the line of client perspectives on how retention in HIV care can be improved. See comment #14 below for additional information on this point.If the authors still think that their work is about cyclical retention, then, this concept should be well defined in the manuscript; we should see findings relating to cyclical retention both in the abstract and the main results section of the paper, and the discussion section should have a central focus on findings relating to the same. At the moment, this is not the case.I did not see any results pertaining to cyclical retention in the abstract, and the conclusion on “continuous differentiated retention support” does not in any way help me to appreciate what the conclusion should have been, if cyclical retention was the main focus of the paper. This conclusion seems to be related to the specific objective on strengthening retention support – than the other two specific objectives that are spelt out on page 6, lines 104-108.Line 90: “LT’s 2wT … developed in 2019 by LT …” There is probably no need for repeating LT.Lines 144-145 (page 9): the authors write: “… to obtain a deeper understanding of the social and behavioral factors that affect clients’ ART engagement in 2wT retention support groups (n=4) and in standard care (n=6)”. This seems to be different from the specific objectives specified on page 6 (client perspectives on retention, suggestions for strengthening retention, and client input on the 2wT mHealth intervention). Reasons for clients’ engagement in the mHealth intervention or standard of care retention activities also seem not to be directly related with the aspect of cyclical retention that is promised in the study title.Lines 146-147, page 9: “We selected focus groups rather than individual interviews to help spur discussion on social norms, values, expectations, and beliefs”. This is the reason provided for using FGDs rather than individual interviews. This reason is not clear to me. How does the study of social norms, values, expectations and beliefs relate to the aspect of cyclical retention that is central to this study? Also, with only FGDs as the data collection method, how did the authors ensure trustworthiness of the data collected or the findings?The authors have done a great job describing how the stratification of their 89 participants was done (see lines 150-161, page 9). However, I have 2 questions here: a) how did the stratification take into consideration the aspect of cyclical retention that is the primary focus of the paper? b) The category of SoC participants included 2 groups who were considered to be “new” on 6-12 months. Can the authors explain why they considered new clients to discuss retention in care?The use of the RE-AIM implementation science framework as the study guiding framework is not clear to me. I don’t see a clear reason/justification for using this framework to guide the study on ‘cyclical retention’. Unless the RE-AIM framework was the model that guided the large study referred to in the manuscript (for which reason, one would have to justify using it for the qualitative component); I am not sure how the RE-AIM framework constructs relate to any of the specific objectives of the study specified on page 6. A little more justification is needed to clearly understand how the RE-AIM framework helped the study team in addressing the three specific objectives as stated on page 6.Qualitative data analysis: I appreciate the use of the rapid qualitative data analysis procedures – but this does not preclude the authors from providing a detailed, step-by-step description of how the data were analyzed. I request the authors to provide a little more detail on how data analysis was executed to help the reader understand what steps were followed, and to ensure that the data analysis was well done; not just a simple summary of issues generated. Results (page 12): The table shown on this page should be given a number. In addition, what does ‘RCT active’ or ‘quasi-experimental active’ mean? These terms are not used in the description of the participant categories on page 9. It would be nice to ensure that there is consistency across the manuscript with regard to these participant categoriesUnless this qualitative study was designed with quantitative aspects in mind, what is the role of the column labeled, ‘M-F Ratio’ on the un-numbered table on page 12? I am not sure if qualitative studies aim to ensure that certain ratios are achieved in the selection of study participants.I have taken a close look at the way the results are organized according to the RE-AIM implementation science framework. Please also refer to comment No. 7 above. The results are organized as follows:

Reach: How best to engage diverse clientsEffectiveness: How to improve … retentionAdoption: Clients’ suggestions to increase retentionImplementation: Clients’ suggestions for improved retentionMaintenance: Client recommendations for maintaining retention gains

Looking at the way the results are organized, I can see that subsection (b)-(d) presents similar results, crafted differently: clients’ suggestions on how to improve retention, and that most, if not all, subsections discuss the “how” of reaching, improving, or maintaining retention gains made. In general, how does this arrangement of results help to answer the primary research question on ‘cyclical’ retention in HIV care?

 I would like to think that this study can take on a different study title, to align with the specific objectives, and that the focus of the paper may change to ‘*client perspectives on improving retention in HIV care among patients receiving treatment in the Lighthouse Trust program in Malawi*’ or something of this nature. This title can well absorb the findings, as reported, especially, if the authors can justify the selection of the RE-AIM implementation science framework as the study guiding framework.

15. Given that there are many subsections used in reporting the results, I would recommend that the authors use a table to summarize the findings according to each of the RE-AIM’s constructs and use short illustrative quotes to support the findings. The current ‘Results’ section is 11 pages long. This section does not need to be this long, double-spacing of text notwithstanding. Besides, there is no reason to double-space the quotations

16. The authors open the discussion with a statement, “… we aimed to describe the perspectives of recipients of HIV care on various retention services offered at LH to gain an understanding on how, when, where, and why clients prefer different retention support services over time”. Again, how does this relate to the study title? Where is the cyclical retention? How does this statement summarize the three specific objectives shown on page 9?

17. The discussion section should be revised to reflect changes made either in the statement of the study title, study objectives, or the reporting of results.

Reviewers' comments:

Reviewer's Responses to Questions

**Comments to the Author**

1. Is the manuscript technically sound, and do the data support the conclusions?

Reviewer #1: Yes

Reviewer #2: Yes

2. Has the statistical analysis been performed appropriately and rigorously? 

Reviewer #1: Yes

Reviewer #2: N/A

3. Have the authors made all data underlying the findings in their manuscript fully available?

Reviewer #1: Yes

Reviewer #2: No

4. Is the manuscript presented in an intelligible fashion and written in standard English?

Reviewer #1: Yes

Reviewer #2: Yes

5. Review Comments to the Author

Reviewer #1: The manuscript is scientifically sound but needs improvement in terminology consistency and alignment with prior literature. Sections like the conclusion are overly long, including unnecessary details that could be omitted.

Reviewer #2: Summary of the Research

This is a great study that sought to describe the clients perspectives regarding the factors affecting ART retention and reengagement and the client’s suggestions regarding retention strategies. To respond to these, Focus Group Discussions were conducted. The study further described the two retention support strategies: the 2 way texting and the Standard of Care support. The FGDs where for clients who were receiving 2 way texting or the Standard of Care support. The RE-AIM theoretical framework was applied to inform the results and findings which was well described.

Among the findings, were the appreciation from the clients for receiving retention support from the Lighthouse Trust – which is commendable. Further findings included a need to address the healthcare providers attitude to ensure a wonderful clinic experience for the clients, addressing the stigma and discrimination, and the need to strengthen the mHealth - 2way texting –retention strategies.

In conclusion, the study acknowledges that Differentiated retention services are key. And a need for retention support investment has been noted. However, It will be appreciated that the authors do put emphasis on recommendations for sustaining these efforts by the ministries of health, especially that we are in the era of sustainability agenda.

It is however noted that the study also sought to provide insight regarding ART clients cycling in and out of care – but, this was not so clear in the manuscript. It will be appreciated if the Authors do show in the form of data how this cycling in and out of care is happening.

Regarding the availability of Data: It is noted that the data is not made publicly available, the authors indicated sensitivity in the data, and can be only be accessed on request.

Detailed Review: Example and Evidence

Title: The title “If he returns, receive him because he has realized that he needs 1 assistance”- gives an impression that this study is about retention for men and boys and providing strategies for retaining men and boys Living with HIV in care. However it is for the general population, both men, boys and women and girls. It will be appreciated if the authors can revisit the title and make it generic, to suit the general population.

The shorter version “Cyclical retention among ART clients in Malawi’’ – may need to also factor the retention support through mHealth, and/or clients perspectives on retention support/strategies. The study is more on the retention in care strategies than the cyclical cascade – it will be appreciated to show from data how the clients are cycling in and out of care

Abstract:

Material and Methods

Line 28 – How many clients were receiving 2way texting and how many were receiving standard of care retention support? What was their duration on ART. What is the average number of clients in each FGD? The study seem to be more on the retention in care than the cyclical cascade. It will be appreciated to show the cyclical cascade – how the cycling in and out of care is happening from data.

Results

Line 32 – it is commendable to learn that the clients appreciate the retention support from the Lighthouse Trust

Line 35 – the recommendation for support for retention in young clients is evident of the need for age disaggregation for the study sample, and to show how they cycle in and out of care.

Conclusion

Line 42 – given the appreciation of the retention support the clients have received, we miss a sentence that recommends strengthening and sustaining such efforts.

Line 43 – it will be appreciated if the authors can make recommendations for ministry of health’s sustainability of the retention support. In the era of the sustainability agenda, it is critical to make reflections of such, because donor funding will not always be fully available.

Introduction

Line 84 – LTFU at 33% in Malawi in 2015 seem too old reference , it will be appreciated to have lates data (at least 5years) to support the concern for high LTFU.

Materials and Methods

Line 110 - Since this is an individual manuscript, it does not seem necessary to make reference of the larger mixed methods study because it is not included in this manuscript. Rather a focus on the qualitative component of the study which this manuscript is all about, and clarity on the qualitative methods that have been applied.

The setting (Lighthouse) and Retention support services (Table 1) at LT, eligibility and Theoretical framework (RE-AIM) have been well described. However in Line 172 it will be appreciated to show what the acronym RE-AIM stands for, prior to describing each component, which has been done well from Line 176.

Line 196 – there is a need to describe the PRISM and how it will be used in the study in translating the results into actions.

Results

Table 2 has not been labelled. We miss age disaggregation in Table 2

There is a need to describe the criteria, for example, clients receiving 2way texting are on Randomized control trial – Active, others are on quasi experimental active or inactive – these concepts have not been described in the manuscript. It will be appreciated if the Authors could clarify these. I have noted that clients that receive the 2way texting are those that have just started ART, so, it will be appreciated to have this clarified on the table.

There is also a need to describe the criteria for retention – the variables that qualifies one to be retained in care – I see there is excellent retention, one would assume that there are categories of retention, and to clarify which is considered as excellent retention, what are the attributes to determine such?. Another consideration would be further the distinguishing variables for those who qualify for 2way texting and the Standard of Care ( shouldn’t there be a reflection on their Viral Load Suppression. It will be appreciated if this could be clarified. Also a need to distinguish between the established and non-established client on ART (See also line 506-508).

Line 226 – it is great and commendable that the clients are happy about the retention support they receive from the Lighthouse Trust and the larger Healthcare system. It is great to learn that in the presenting of the results the Authors included these appreciation comments from the clients. Also noted in Line 335- 357 the reflection on areas for improvement for the healthcare providers.

Line 269 – the recommendation for support for retention in young clients is evident of the need for age disaggregation for the study sample, and to show how they cycle in and out of care. The same goes for older clients.

Discussion

Line 506-508 – There is also a need to distinguish between the established and non-established client on ART in the results section.

Line 551 – introduces Recipients of care – The study has been using clients throughout, so consistency will be appreciated, or an indication that clients and recipients of care will be used alternatively.

Line 561 – the two categories of retention are discussed here, a need to have them described in the results section as already suggested, the same goes for new on ART and established on ART.

Conclusion

Line 579 – we miss recommendations for sustainability. It is clear in Line 578 that LT is unable to assume greater retention cost. But in the era of Sustainability agenda, there is a need to put forth recommendations for organizations sustainability for such retention efforts.

6. PLOS authors have the option to publish the peer review history of their article (what does this mean? ). If published, this will include your full peer review and any attached files.

**Do you want your identity to be public for this peer review?** For information about this choice, including consent withdrawal, please see our Privacy Policy .

Reviewer #1: No

Reviewer #2: **Yes: ** Lufuno Malala

---

## [Author Response · Author response to Decision Letter 1]

13 Jan 2025

1.1 The title is very long but lacks the necessary information. A good title should present the major area, study participants, site, and study design.

Thank you for this suggestion, we’ve included additional detail in the title to better reflect design and major area of study.

1.2 The article’s introduction is lengthy; a more concise version would be helpful since the details are repeated in the methods section.

Thank you for this suggestion. We’ve reworked the introduction to be more concise and less duplicative of the methods section.

1.3 Lines 18 and 54: This manuscript focuses on retention in HIV care, but the term “retention in ART” is used in some places. The authors should consistently use “retention in HIV care,” which accurately reflects the phenomenon.

Thank you for this suggestion. While retention in HIV care is more encompassing, it is retention on ART that we are interested in as a key and well known indicator of HIV treatment success. ART retention is critical for achieving viral suppression. We therefore prefer to keep this terminology, see also (https://www.who.int/teams/global-hiv-hepatitis-and-stis-programmes/hiv/treatment/service-delivery-adherence-retention#:~:text=building%20and%20mentoring.-,Adherence%20and%20retention,as%20well%20as%20treatment%20monitoring.). Additionally, retention on ART is the preferred term used by the Lighthouse Trust. As such, we have not changed retention on ART to retention in HIV care throughout the manuscript but did mention HIV care where we felt this was clearer.

2.1 Consider revising the headings, such as instead of materials and methods only using methods.

Thank you for this suggestion, we used manuscript organization guidance outlined on the PLOS ONE website which includes “Materials and Methods” as a heading (https://journals.plos.org/plosone/s/submission-guidelines#loc-manuscript-organization).

2.2 Consider making the writing more succinct by removing redundant sentences.

Thank you for this feedback. We’ve reorganized the introduction and discussion sections to make the writing more succinct.

2.3 Line 22, objectives are unclear as they don’t elaborate on exploring mHealth intervention. The objectives should state that we aimed to explore participants’ experiences for improved retention in care using 2-way texting retention support.

Thank you for this suggestion, we’ve revised the objective statement in the abstract.

2.4 Line 19 – 20, “Retention is not static—gaining traction can be removed.

Thank you for this suggestion, we’ve revised the abstract but kept sentences around mHealth and retention over time as they are key providing background on the study’s objectives.

2.5 Line 26, “on ART,” is incomplete; it should read like on retention in care.

See 1.3 above.

2.6 Line 29, “We used the Re-Aim” should be before line 28, “Rapid qualitative.”

Thank you for this suggestion, we have incorporated this change into the revised version of the manuscript.

2.7 Lines 31 – 39 are like the conclusion; results should summarize the themes and subtheme from which conclusions can be drawn.

Thank you for this suggestion. We’ve reorganized the abstract with results summarizing main themes.

2.8 Line 56. Consider using the terms adherence to ART and retention in HIV.

See 1.3 above.

2.9 Line 473-487. This section requires organization and is more focused on summarizing what was explored and what was found. The framework should be discussed earlier, not in the last sentence.

Thank you for this suggestion, we’ve reorganized and shortened the discussion section as well as included the RE-AIM framework earlier.

2.10 Consider making this section more relatable to the study. Line 575 – line 580 can be in the future directions.

Thank you for this suggestion, we’ve rewritten the conclusions to be more relatable to the study.

2.11 All websites cited should also have a date of access.

Thank you for identifying this, all cited websites now have a date of access.

---

## [Decision Letter · Decision Letter 1]

19 Mar 2025

PONE-D-24-35611R1“If he returns, receive him because he has realized that he needs assistance”: a qualitative study exploring preferences for retention on antiretroviral therapy support in Malawi among Lighthouse Clinic clientsPLOS ONE

Dear Dr. O’Bryan,

Thank you for submitting your manuscript to PLOS ONE. After careful consideration, we feel that it has merit but does not fully meet PLOS ONE’s publication criteria as it currently stands. Therefore, we invite you to submit a revised version of the manuscript that addresses the points raised during the review process.

We look forward to receiving your revised manuscript.

Kind regards,

Joseph KB Matovu, Ph.D.

Academic Editor

PLOS ONE

Journal Requirements:

Additional Editor Comments:

The authors have addressed the main comments from the reviewers. They should also address the minor comments from one of the reviewers.

Reviewers' comments:

Reviewer's Responses to Questions

**Comments to the Author**

1. If the authors have adequately addressed your comments raised in a previous round of review and you feel that this manuscript is now acceptable for publication, you may indicate that here to bypass the “Comments to the Author” section, enter your conflict of interest statement in the “Confidential to Editor” section, and submit your "Accept" recommendation.

Reviewer #1: All comments have been addressed

Reviewer #2: All comments have been addressed

2. Is the manuscript technically sound, and do the data support the conclusions?

Reviewer #1: Yes

Reviewer #2: Yes

3. Has the statistical analysis been performed appropriately and rigorously? 

Reviewer #1: Yes

Reviewer #2: N/A

4. Have the authors made all data underlying the findings in their manuscript fully available?

Reviewer #1: Yes

Reviewer #2: Yes

5. Is the manuscript presented in an intelligible fashion and written in standard English?

Reviewer #1: Yes

Reviewer #2: Yes

6. Review Comments to the Author

Reviewer #1: The manuscript has been revised and is a valuable contribution to research. Previous concerns have been addressed, and the authors have clearly presented the intervention's development and participants' perspectives. This transparency will help identify potential shortcomings and guide implementation. In my view, all intervention trials should be preceded by a qualitative phase of development.

Reviewer #2: This is an important study, addressing all the key interventions to ensuring retention in care for clients on ART. It is appreciated that the authors addressed all the reviewer's comments - including defining retention in care, established on ART and addressing the recommendations for sustainability.

However, Just few considerations still outstanding:

Line 75-76: It is not clear if there is no latest data available to talk to the LTFU rates in Malawi apart from the 33% in 2015. It will be good to have the latest data.

Just to note: The table under the results section line 192 , still does not have a title/heading

7. PLOS authors have the option to publish the peer review history of their article (what does this mean? ). If published, this will include your full peer review and any attached files.

**Do you want your identity to be public for this peer review?** For information about this choice, including consent withdrawal, please see our Privacy Policy .

Reviewer #1: No

Reviewer #2: **Yes: ** Lufuno Malala

---

## [Author Response · Author response to Decision Letter 2]

20 Mar 2025

Line 75-76: It is not clear if there is no latest data available to talk to the LTFU rates in Malawi apart from the 33% in 2015. It will be good to have the latest data.

Thank you for this suggestion, we’ve revised the sentence with a 2022 updated estimate for retention on ART at 12 months after initiation.

Just to note: the table under the results section line 192, still does not have a title/heading.

Thank you for identifying this omission, a title has been added to the table (table 2) in the results section.

---

## [Editor Report · Decision Letter 2]

23 Apr 2025

“If he returns, receive him because he has realized that he needs assistance”: a qualitative study exploring preferences for retention on antiretroviral therapy support in Malawi among Lighthouse Clinic clients

PONE-D-24-35611R2

Dear Dr. O’Bryan,

We’re pleased to inform you that your manuscript has been judged scientifically suitable for publication and will be formally accepted for publication once it meets all outstanding technical requirements.

Kind regards,

Joseph KB Matovu, Ph.D.

Academic Editor

PLOS ONE
---

## [Editor Report · Acceptance letter]

PONE-D-24-35611R2

PLOS ONE

Dear Dr. O’Bryan,

I'm pleased to inform you that your manuscript has been deemed suitable for publication in PLOS ONE. Congratulations! Your manuscript is now being handed over to our production team.

Kind regards,

on behalf of

Dr. Joseph KB Matovu

Academic Editor

PLOS ONE